# Large influence of dust on the Precambrian climate

Peng Liu [1], Yonggang Liu [1✉], Yiran Peng[2], Jean-François Lamarque [3], Mingxing Wang [2] & Yongyun Hu [1✉]

On present-day Earth, dust emissions are restricted only to a few desert regions mainly due to the distribution of land vegetation. The atmospheric dust loading is thus relatively small and has a slight cooling effect on the surface climate. For the Precambrian (before ~540 Ma), however, dust emission might be much more widespread since land vegetation was absent. Here, our simulations using an Earth system model (CESM1.2.2) demonstrate that the global dust emission during that time might be an order of magnitude larger than that of the present day, and could have cooled the global climate by ~10 °C. Similarly, the dust deposition in the ocean, an important source of nutrition for the marine ecosystem, was also increased by a factor of ~10. Therefore, dust was a critical component of the early Earth system, and should always be considered when studying the climate and biogeochemistry of the Precambrian.

[1] Department of Atmospheric and Oceanic Sciences, School of Physics, Peking University, 100871 Beijing, China. [2] Department of Earth System Science, Tsinghua University, 100084 Beijing, China. [3] National Center for Atmospheric Research, Boulder, CO 80305, USA. ✉email: ygliu@pku.edu.cn; yyhu@pku.edu.cn

The earliest land plants probably appeared between late Cryogenian (720–635 Ma) and middle Cambrian (541–485 Ma)[1,2]. Therefore, land surfaces were bare with no vegetation throughout most part of Earth's history. Without vegetation coverage on land, surface dust emission was likely much more widespread and intense than that at present-day Earth. Atmospheric dust loading would be much higher and have larger impacts on the climate; dust deposition into oceans could provide a large amount of nutrients to the marine ecosystem (i.e., the iron fertilization effect[3]); it could also be the mechanism for the formation of widespread tabular sediment bedding sometimes observed in the Precambrian[4]. Despite of its importance, the dynamic characteristics of dust during this period have never been investigated seriously.

Dust particles can affect climate directly by absorbing and scattering solar radiation[5–7], and indirectly by modifying the properties and lifetime of clouds[8,9]. At present, the global mean solar radiation received at the surface is reduced by approximately −1 to −3 W m$^{-2}$ due to the direct radiative effect of dust[10,11]. The indirect effect is also important, but the quantitative estimate of this effect is difficult[12]. The global mean indirect effect at the surface has been estimated to be around −2.05 W m$^{-2}$ [13]. About 1000–3000 trillion grams (Tg) of dust emissions are injected into the atmosphere each year in the present day[14]. If such dust sources were diminished, for example, during the mid-Holocene (9–6 ka, ka = 10$^3$ years ago), the African monsoon and El Nino Southern Oscillation (ENSO) would be significantly impacted[15,16], and the global mean sea surface temperature could increase by as much as 0.3 °C[17]. So far, the influence of dust was omitted in almost all modeling studies of the Precambrian climate[18–21]. This is probably because of the relatively weak global impact of its modern counterpart as described above. Its importance has only been considered in the melting of a Neoproterozoic snowball Earth[22] that occurred during the Cryogenian[23,24]. In both of these studies, however, dust had a warming effect either by lowering the surface albedo by polluting the surface ice[24] or lowering the planetary albedo by floating in the air above glaciated surfaces[23].

Here, we examine the dust cycle during the Precambrian as well as its climate effects. To do this, a series of climate simulations are carried out using an Earth system model, CESM1.2.2, and a Precambrian continental configuration (see "Method") for a few different surface erodibilities. Modeling results show that both atmospheric dust loading and dust deposition in the oceans were many times larger than those of present day. This high dust loading could cool the global climate by ~10 °C in a relatively warm background climate, much larger than its modern counterpart. The role of dust in Precambrian climate could be as important as that on present-day Mars[25], and should always be included when studying the Precambrian climate of Earth.

## Results

### Emissions and depositions under different erodibilities. In the absence of vegetation, the surface emission of dust is mainly determined by the surface erodibility ($k$), wind stress, and soil wetness. The surface erodibility is highly variable on the present-day land, most likely proportional to the upstream runoff collection area[26]. It varies from 0.0 to 5.7, with a global mean value of 0.15. Since the spatial distribution of $k$ for the Precambrian is unknown, it is assumed to be uniform globally but a series of four values, 0.0375, 0.075, 0.15, and 0.3, are tested. For the present-day Earth, a uniform $k$ produces similar spatial pattern and magnitudes of dust emission to those produced from a spatially varying $k$[26]. This justifies the use of a uniform $k$ herein to some extent.

Compared to the case where modern types of vegetation are assumed to be able to grow (computed by the dynamic vegetation

model of CLM4; Supplementary Fig. 1), the dust emission is indeed much larger in the case where there is no land vegetation (Fig. 1). The regions of dust emission have significant seasonal variations and are mainly located in the subtropical zone where the soil is dry; the seasonal shift of wind is also important (Supplementary Fig. 2). Dust emission from high latitudes is suppressed by land snow cover (Supplementary Fig. 3). Even for the case with the smallest $k$, i.e., 0.0375, the total surface emissions and atmospheric dust loading are more than 15 and 10 times higher than the present-day values, respectively (Fig. 2a and Supplementary Table 1). When vegetation is present, they are only 5.7 and 4.8 times that of present-day, respectively (Supplementary Table 1). Note that the continental configuration employed herein is appropriate for the Cryogenian[27]; the total area of the continents is only ~110 million km$^2$, ~73% that of the present-day.

The total global dust emission (no vegetation) scales almost linearly with surface erodibility, increasing from 36,463 Tg yr$^{-1}$ when $k$ is 0.0375–132,303 Tg yr$^{-1}$ when it is 0.3 (Fig. 2a and Supplementary Table 1). The increasing rate is approximately a factor of 1.53 per doubling of $k$. The same is true for atmospheric dust loading, which increases from 312 Tg to 1114 Tg (Supplementary Table 1 and Supplementary Fig. 4). As $k$ increases, wind stress decreases, but land precipitation also decreases significantly, which partially compensates the effect of wind stress weakening (Supplementary Fig. 5). Moreover, the fraction of fine dust particles (0–2.5 μm) in the total atmospheric dust loading increases from 45.8% when $k$ is 0.0375 to 56.5% when $k$ is 0.3 (Supplementary Table 1).

After being blown into the atmosphere by wind, dust particles return to the surface by dry and wet depositional processes. Due to decreasing precipitation with increasing $k$ (Supplementary Fig. 5), the fraction of wet deposition of dust decreases slightly from 32.5% when $k$ is 0.0375 to 30.0% when $k$ is 0.3. For all the experiments without vegetation, roughly one-third of the total emitted dust is deposited into the oceans (Supplementary Table 1), ~10–35 times that of present day. This means that the dust transportation might have been a very important way of providing nutrients to the oceanic ecosystem in Precambrian, and might be important for life evolution[3]. Moreover, the dust depositional rate at some coastal regions is equivalent to a sedimentation rate of ~10 mm kyr$^{-1}$ when $k$ is 0.0375 (Supplementary Fig. 6), similar to that along the coast of North Africa during the past 2000 years[28]. This is enough to create a thick tabular sediment bedding in a few million years, like that observed in Montana, United States, as proposed recently in ref. [29]. Such sediment bedding has often been interpreted as due to sheet flooding[4].

**Impact on climate**. Globally, 22.5 W m$^{-2}$ less sunlight is received at the surface when $k = 0.0375$ (no vegetation) compared to the case when there is no dust ($k = 0$), and the global mean surface temperature is reduced by 11.2 °C. It would be reduced by another ~9 °C if the $k$ is increased to 0.3, ~3 °C per doubling of the erodibility (Fig. 2c and Supplementary Table 1). Correspondingly, the sea ice edges expand from ~55° latitude when there is no dust to the tropical region when $k = 0.3$ (Supplementary Fig. 7). The precipitation decreases globally as $k$ increases (Supplementary Fig. 5), with the global mean precipitation decreasing from 2.9 mm day$^{-1}$ when $k = 0$–1.0 mm day$^{-1}$ when $k = 0.3$.

For each doubling of $k$, ~12 W m$^{-2}$ more sunlight, which would reach the surface, is blocked by dust (Fig. 2b and Supplementary Table 1), but the surface temperature decreases by only about 3 °C. This is primarily because the dust absorbs sunlight directly and heats the atmosphere, which in turn heats the surface. When $k$

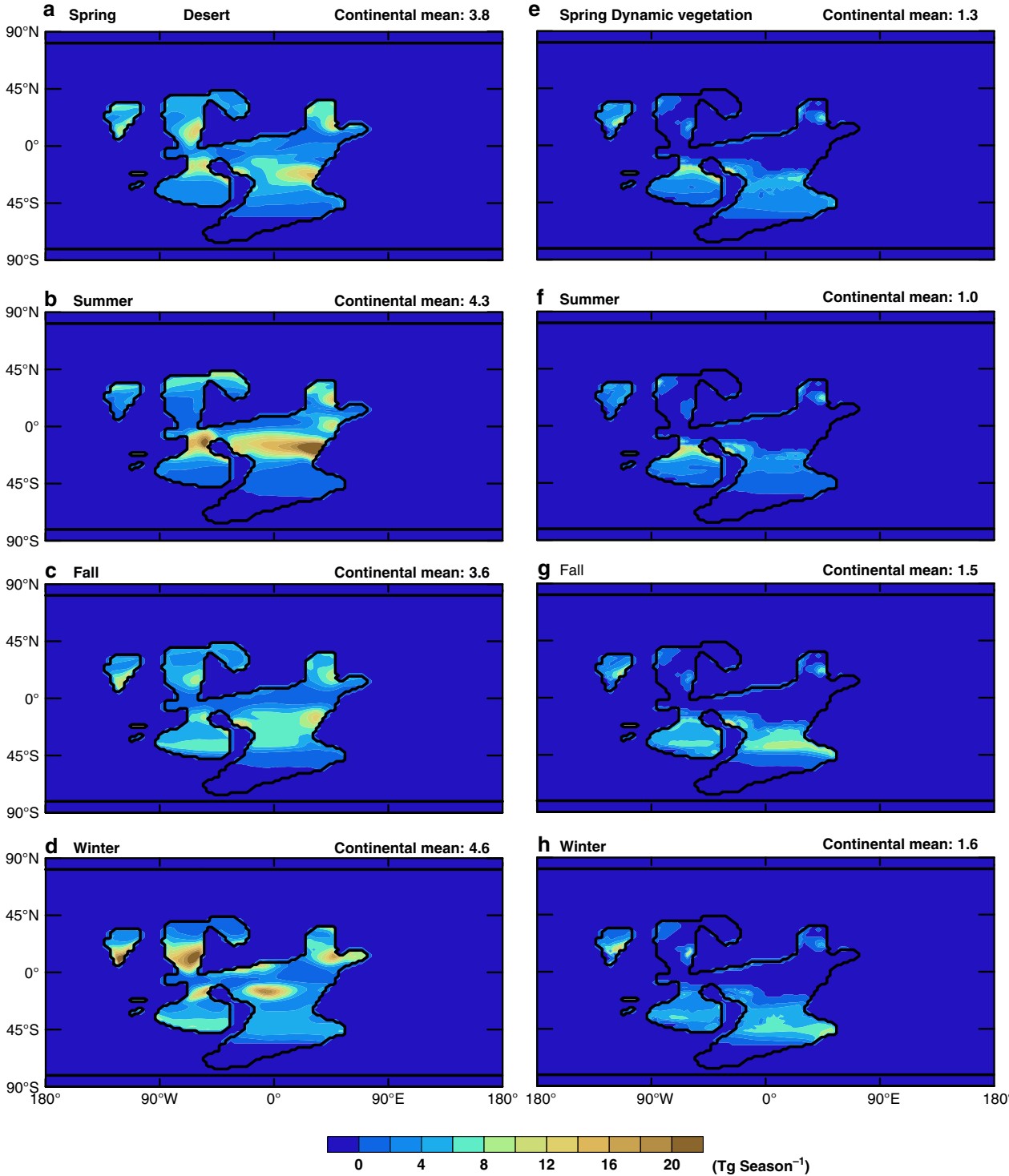

**Fig. 1 Seasonal dust emission simulated by the model.** Dust emission for four seasons when (**a–d**) vegetation is prescribed to be absent on land and (**e–h**) vegetation is simulated by the model. The surface erodibility is spatially and temporally uniform with a value of 0.0375. Each season has three consecutive months and the Spring starts from March. Black contour lines represent continental boundaries. Values on the upper right corner of each panel are the average of dust emission over the continents. Unit: Tg season$^{-1}$.

increases, the atmospheric heating rate due to dust increases in the troposphere, with the most significant increase occurring in the lower troposphere (Fig. 2e). The temperature of the mid- to upper-troposphere thus remains almost unchanged while the surface is being cooled (Fig. 2f); the temperature profiles deviate from a moist adiabat as obtained when $k$ is 0 (compare the curves

with non-zero $k$ to the curve with $k = 0$ in Fig. 2f). In the most extreme case tested here, i.e., $k = 0.3$, the global mean atmospheric temperature profile is inversed near the surface (the dark blue curve in Fig. 2f). Using a 1-D radiative-convective model, Farrell and Abbot[30] also found that an inversion developed when dust emission was increased to more than 30 times the present-day

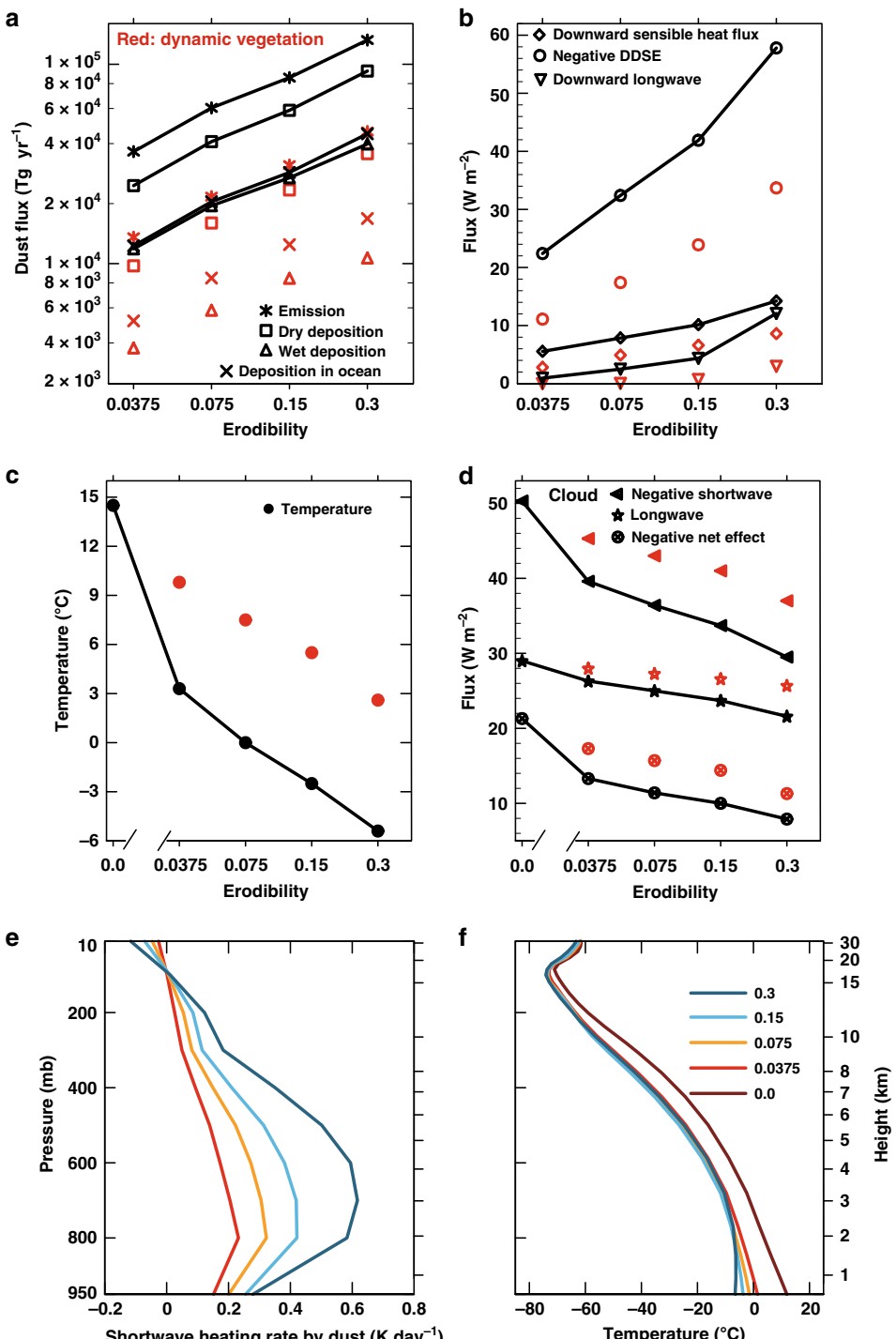

**Fig. 2 Variations of global dust, radiative forcings and temperature with surface erodibility.** All variables are in global mean except in (**a**) where the global sum is shown. **a** The surface emission, dry and wet deposition onto the surface, and total (dry + wet) deposition into the ocean, of dust every year (unit: Tg yr⁻¹); **b** Dust Direct Shortwave Effect (DDSE) which represents the reduction of solar radiation received at the surface, anomalous downward sensible heat flux and longwave radiative fluxes at the surface due to dust (see method section for how they are diagnosed); **c** Annual mean surface temperature; **d** Longwave, shortwave and net cloud radiative forcings, where the latter two are negative but their absolute values are shown for the sake of convenience; **e** Heating rate of the atmosphere by dust; **f** Vertical temperature profiles of the atmosphere. The red markers in (**a–d**) are values for the cases with dynamic vegetation turned on. Note that both the horizontal and vertical axes in (**a**) are in base-10 logarithmic scale, so the lines in this panel indicate linear relationship. The x-axes for erodibility (dimensionless variable) in panels (**b–d**) are in logarithmic scale except the zero erodibility point.

level, and the deep convection was completely shut off due to the inversion. The increasing fraction of fine dust particles with increasing $k$ (Supplementary Table 1) is likely due to the gradual weakening of the deep convection, which is responsible for

removing a large part of the fine particles residing in the upper and middle atmosphere. Such temperature profiles affect the surface energy budget in three means, all of which compensate the cooling due to direct blocking of sunlight by dust.

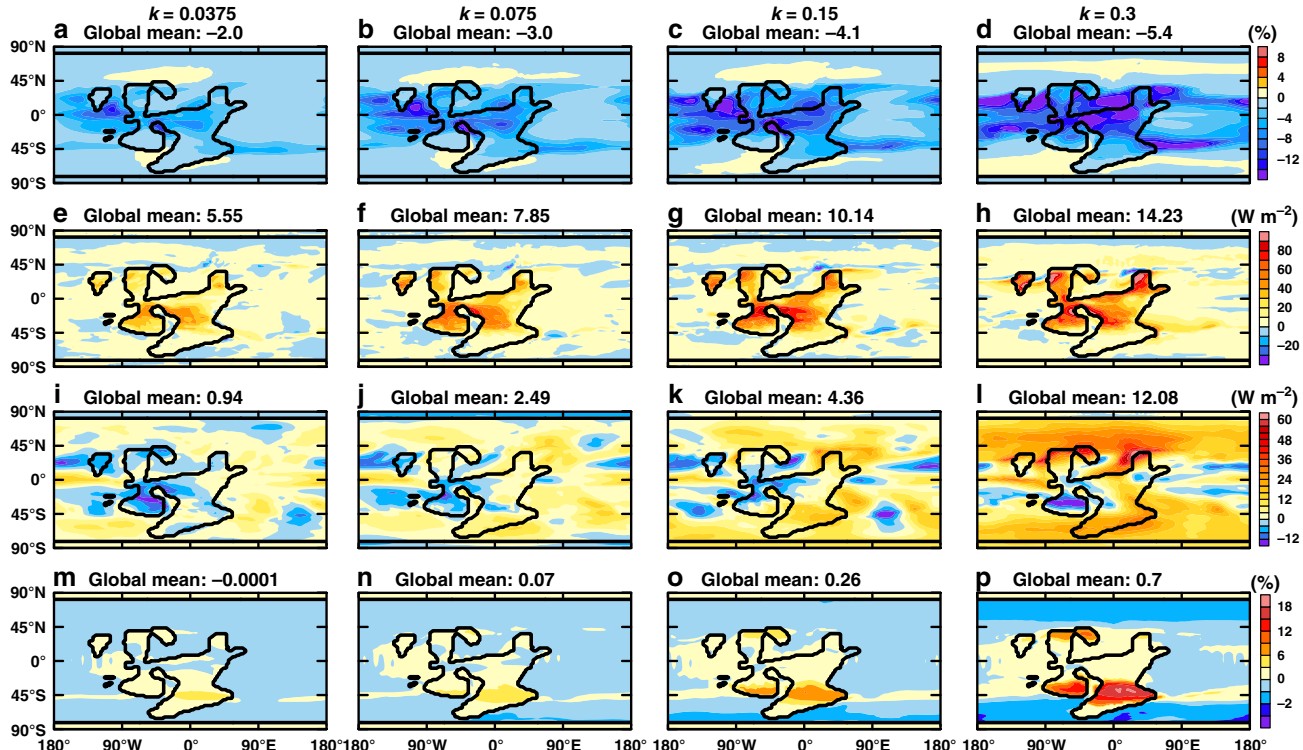

**Fig. 3 Warming effect of the dust on the surface. a–d** Change of cloud albedo due to dust; **e–h** The impact of dust on surface sensible heat fluxes, positive is downward; **i–l** The impact of dust on downward longwave radiative fluxes, positive downward; **m–p** The change of surface albedo due to dust. Each column shows the results for a specific erodibility as labeled on the top. Black contour lines represent continental boundaries. The number in the upper left corner of each panel is the global averaged value. See the Methods section for how changes in cloud albedo, surface sensible heat fluxes, downward longwave radiative fluxes, and surface albedo due to dust are diagnosed.

The first effect is that the stability of the atmosphere is greatly enhanced and the convective activities inhibited; the cloud albedo is substantially reduced (Fig. 3a–d), causing a net warming effect to the surface climate (Fig. 2d). In fact, the planetary albedo increases only slightly from 0.383 to 0.405 when $k$ is increased from 0.0375 to 0.3 (Supplementary Fig. 8), despite a substantial increase of atmospheric dust loading (Supplementary Table 1). The second effect is that the heat loss from the ground to the atmosphere through sensible heat flux is reduced (Fig. 3e–h). The third effect is that the downward longwave radiative flux from the atmosphere to the ground is increased (Fig. 3i–l).

There is a fourth warming effect to the surface, unrelated to the atmosphere; deposition of dust reduces the albedo of ice and snow. The albedo of sea ice can be reduced by as much as 3% when $k = 0.3$ (Fig. 3m–p), mainly because the snow on sea ice evaporates away easily resulting in thinner snow depth (Supplementary Fig. 9). However, this effect is significant only when the climate is cold enough such that there exists large areas of sea ice and the snow precipitation rate is low (so that dust concentration in snow is high). This warming effect is clearly overwhelmed by the expansion of sea ice when $k$ is increased from 0 (Supplementary Fig. 8a) to 00375 (Supplementary Fig. 8b), because the former climate is still quite warm (Fig. 2c).

Nevertheless, dust has an overall strong cooling effect (>10 °C) to the surface climate when there was no land vegetation, even for a small $k$. When dynamic vegetation is turned on, the global mean surface temperature increases by more than 6.5 °C in all cases (Fig. 2c), hence the evolution of land vegetation has probably greatly suppressed dust emission and warmed up the climate (for fixed solar constant and greenhouse gas concentrations). The effect of vegetation as simulated herein might have been underestimated because the employed land model is known to underestimate tundra vegetation coverage[31]. This caveat may be exaggerated in the overall cold climates when dust is present (Supplementary Fig. 7). Moreover, the duration of the simulations is too short for vegetation to fully develop. In a test under pre-industrial conditions, the extent of vegetation calculated by the model is much smaller than that observed, even after 2600 years (Supplementary Fig. 10). Therefore, the difference in global mean surface temperature between the cases with and without dynamic vegetation as presented in Fig. 2c is likely significantly underestimated. The climates of the former should be much warmer if the simulations were carried out for longer and the biases in vegetation area were absent. The simulations with dynamic vegetation were not carried out further because of computational cost, but are long enough to demonstrate the point that dust could induce substantial cooling when vegetation is absent.

## Discussion

Because the continental configuration used here is most appropriate for the late Neoproterozoic (800 Ma-541 Ma;[32]), the results described above indicate that the Neoproterozoic climate (when not in or near a snowball state) was much colder than those simulated in previous studies (e.g.,[18,33,34]) where effects of dust were not considered. We also argue that large atmospheric dust loading should be a universal feature before land vegetation developed, and that the results here can be reasonably generalized to other continental configurations during the Precambrian. That means, the effect of dust on deep-time paleo-climate was enormous and should always be considered when simulating early Earth climatic conditions as long as large area of continents have formed. Recent studies showed that the area of continents had probably not been much smaller than that of present day since 2.5

billion years ago[35,36]. Whether dust has promoted the initiation of Neoproterozoic snowball Earth or not is uncertain because it can lower the surface albedo and warm the climate when there is a large amount of sea ice (such as in a near snowball state); this effect may overwhelm its cooling effect and warrants further investigation.

Soil crust may form on the surface of bare soil; it consists of soil particles associated either by physical compaction or biological bodies (cyanobacteria during the Precambrian). The biogenic soil crust can be especially strong if the crust is thick and the concentration of organic material in the crust is high. It can thus reduce the dust emission by wind. It develops when the region is not hyperarid and little disturbed by wind or water flow[37]. Soil crust should be present during the Precambrian where wind is weak and precipitation is moderate or heavy, but its detailed distribution is unknown. Given the fact that strong surface dust emission and deposition always occurs in present day and during the Earth's history (e.g.,[38]), we think it might not be a key factor in controlling global dust emission and jeopardizing the major conclusions herein. Soil crust generation by cyanobacteria is not considered in the current generation of climate models, and the uncertainty it induces has hopefully been contained crudely in the range of surface erodibility ($k$) we have tested. In the future, it will be worthwhile to model the dynamics of soil crust explicitly and its influence on global dust emission and dust loading during the Precambrian.

A much larger dust loading than the present-day value is not unimaginable since it increased by approximately a factor of 2 during the last glacial maximum (~21 ka)[39], when most of vegetation was still present and two additional large ice sheets covered the high-latitude regions. The largest uncertainty of the present study is in the value of surface erodibility $k$. According to the linear relationship between global mean surface temperature and $k$, the global cooling by dust could reach ~5 °C even if $k$ is four times smaller than the smallest value used here. This is without considering the possible extra cooling due to indirect effect of dust. Due to the uncertainty in $k$, we may argue that dust could cause significant global cooling of the Precambrian climate, with an approximate value of 10 °C.

The source of dust, i.e., $k$, should be a function of time; it may vary with glacial cycles since the continental glaciations are a powerful mechanism for producing fine sediment particles[39]. Therefore, post-snowball Earth time might be characterized by especially dusty conditions, and the deposition of dust in the ocean might induce strong primary production and organic carbon burial. The latter would have caused atmospheric oxygen to rise as was observed (although the uncertainty in dating prevents us to know exactly when oxygen rose)[40].

In summary, model simulations indicate that dust emission and atmospheric dust loading on Earth before vegetation appeared on land might have been ~10 times higher than that on present-day Earth. The corresponding high dust deposition could have produced widespread tabular sediment beddings observed at various locations, and could have been an important nutrition source for ocean life. The high atmospheric dust loading could have had a strong cooling effect (~10 °C) on the climate of early Earth, and might have an effect on atmospheric chemistry as well. Therefore, future studies on early Earth climate, atmospheric chemistry, and biogeochemical cycle etc. should always consider the influence of dust.

## Methods

**Model configuration**. The CESM 1.2.2 can simulate the dynamic and thermo-dynamic processes of the atmosphere, land, sea ice, oceans, land ice, and river systems[41]. For the purpose of the current study, the land ice component is not turned on. The atmospheric component has two versions: Community Atmosphere Model version 4 (CAM4) and CAM5. The former is used herein because the latter can only be run with its chemical package fully turned on[42], which is computationally very expensive[43]. CAM4 is run with its finite volume dynamical core[44]. The land component, Community Land Model version 4 (CLM4), includes processes associated with snow, water storage, vegetation, and dust emission, etc.[45,46]. All simulations are run with f19_g16 configuration in which CAM4 and CLM4 have a horizontal resolution of $1.9° \times 2.5°$, the ocean component (Parallel Ocean Program Ocean model version 2, POP2)[47] and ice component (Community Ice CodE version 4.0, CICE4)[48] have a normal horizontal resolution of ~1° ($384 \times 320$ grid points in the meridional and zonal directions, respectively). The atmosphere and ocean have 26 and 60 layers in the vertical direction, respectively.

CAM4 calculates dust emission, transport, and dust radiative effect according to several key physics parameterizations, i.e., soil erodibility, dust emission size distributions, wet deposition, and optical properties[11,49]. The soil erodibility ($k$) represents the efficiency of soil in producing dust aerosols under a given wind stress[26,50]. It is usually prescribed in an erodibility map[51,52]. The global mean value of $k$ is 0.15 for the present-day Earth in this model. The dust emission scheme used is Dust Entrainment And Deposition (DEAD) scheme, which is based on saltation-sandblasting process related to wind friction velocity (here we pick 0.35 as the dust emission factor as CAM5), soil moisture, and vegetation/snow cover[53]. The Bulk Aerosol Model scheme in CAM4 calculates dust emission, transportation and deposition for four different size bins (0.1–1.0, 1.0–2.5, 2.5–5.0, and 5.0–10 μm in diameter)[49,54], the same as those for sulfate, sea salt, organic carbon, and black carbon[55].

CAM4 considers only the shortwave radiative effect of dust, while neglecting its longwave radiative effect and its effect as cloud condensation nuclei[11,56]. Previous studies indicated that the longwave radiative effect due to dust was much smaller than the shortwave effect both at the surface[12] and the top of atmosphere[11]. The indirect effect of dust, i.e., its interaction with clouds, is difficult to estimate but could be important for local climate[57,58]. Dust can also affect climate by lowering the albedo of snow and ice[59]. This effect is especially important in a cold climate where the total snow and ice area is large. This effect is simulated in CLM4 for land snow[45] and CICE4 for sea ice and snow on it[41].

The land component, CLM4, can be run with the dynamic vegetation scheme (CNDV) which includes the carbon and nitrogen cycle simulation[60]. The development of unmanaged plant functional types (i.e., tree, grass, and shrub) is based on the warmest minimum monthly air temperature, minimum precipitation (>100 mm yr⁻¹), and minimum annual growing degree-days above 5 °C[60]. CNDV simulates a reasonable present-day vegetation distribution but underestimates tundra vegetation cover and overestimates tree cover[31].

**Experimental setup**. The continental configuration for 720 Ma constructed by Li et al.[27] is employed here. Lacking information on surface topography, the continents are assumed to be almost flat with an average elevation of about 400 m above sea level, which is close to the global mean surface elevation of present-day Earth. The elevation is assumed to be highest (450 m) at the center of the global continents as a whole, and decreases linearly toward the edges so that the rivers are directed outwards. The depth of the ocean is assumed to be 4000 m, with an idealized mid-ocean ridge added on the ocean floor to improve the convergence of the ocean model[61]. The solar constant is set to be 94%$S_0$, where $S_0$ is the present-day value (~1367 W m⁻²), since the Sun was dimmer in the past[62]. The orbit of the Earth is assumed to be the same as that of year 1990, which is the default setting of the model. These configurations are similar to those in Liu et al.[33,61] where the initiation of snowball Earth events were studied.

To ensure that the climate obtained is not in a snowball Earth state, $CO_2$ mixing ratio is set to a relatively high value of 2000 ppmv. The levels of $CH_4$ and $N_2O$ are the same as pre-industrial values (805.6 and 276.7 ppbv, respectively), and the aerosols such as black carbon, sulfate and organic carbon are omitted. In the control run, $k$ is set to 0.15 everywhere, which is the global mean of the present-day value. After the control run reaches equilibrium state (after 2300 model years), four additional runs are branched off in which $k$ is modified to 0.3, 0.075, 0.0375, and 0.0 (no dust), respectively (Supplementary Table 1). All the branch runs are continued for at least 400 years so that the global mean net radiative flux at the top of atmosphere is <±0.3 W m⁻².

When the simulations with non-zero $k$ reach statistical equilibrium, one run is branched off for each case in which the dynamic vegetation scheme (CNDV) is turned on. These runs are continued for 700 years so that vegetation and climate reach new equilibrium. The changes of global fraction of vegetated land cover are small (~0.5% century⁻¹) by the end of simulations and the global mean net radiative at TOA is again <±0.3 W m⁻². The last 100 years of data for each run are averaged for the analyses presented in the main text.

Diagnosis of the dust radiative impact. We take a similar approach as in[63] to estimate the direct shortwave radiative effect of dust. In each time step, the radiative fluxes are calculated twice, one with the radiative function of dust turned off and the other turned on. Both radiative fluxes are output and the difference between them gives us the radiative effect of dust. Five years of simulations are carried out for each case for this purpose. In this method, the radiative effect is diagnosed without affecting the climate and is thus precise. The DDSE shown in Fig. 2b is calculated this way.

The longwave radiative effect due to dust is not considered by the model and thus cannot be diagnosed in the same way as above in CAM4. However, since dust absorbs solar radiation to warm ambient air parcels, the warm atmosphere does heat the surface by downward longwave radiation. To diagnose this, we carried out two short runs for each case, one with dust and the other without. The durations of runs are only 1 year to minimize the drift of climate. The anomalous longwave radiative flux and downward sensible heat flux in Fig. 2b, and all the anomalies in Fig. 3 are obtained by calculating the difference in respective variables between the two runs. Because the climate does drift a little during the run when dust is removed, the diagnosed radiative effect is less precise than that diagnosed using the method above, but we found that the shortwave radiative effect of dust diagnosed by these two methods are comparable.

## Data availability
The model results that support the findings of this study are available on Zenodo with the identifier https://doi.org/10.5281/zenodo.3823422.

## Code availability
CESM is open-source climate model and available at http://www.cesm.ucar.edu/models/cesm1.2/.

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

## Acknowledgements

We thank Yosef Ashkenazy and Jayne Belnap for the discussion on biological soil crust. Y.L., P.L., and Y.H. are supported by the National Natural Science Foundation of China under grants 41875090, 41888101 and 41761144072. Y.P. and M.W. are supported by the National Natural Science Foundation of China under grant 41775137. The computation was supported by the High-performance Computing Platform of Peking University. The CESM project is supported primarily by the National Science Foundation (NSF). The National Center for Atmospheric Research (NCAR) is a major facility sponsored by the NSF under Cooperative Agreement 1852977.

## Author contributions

Y.L. and Y.H. proposed the project, designed the numerical experiments. Y.L. set up the first run, P.L. performed most of the simulations. Y.P., J.L. and M.W. helped set up the model and diagnose the radiative fluxes. Y.L. and P.L. developed the paper together. All authors contributed to discussion and paper revision.

## Competing interests

The authors declare no competing interests.
