## [Peer Review File · Nature Communications]

Reviewer #1 (Remarks to the Author):

Paper: Large Influence of Dust on the Precambrian Climate

Authors: Liu et al.

Reviewer: Dorian S. Abbot

Date: March 2, 2020

Overview: This paper points out the fact that many previous paleoclimate modeling efforts have neglected the impact of aerosol dust. More dust aerosol generally leads to cooling, although there are some compensating warming effects of dust described in the paper. The paper describes GCM calculations that show that the cooling due to dust could be as much as 10 K in the global mean. The effect of dust should be greatest before land plants evolved, which makes this work most relevant to the Precambrian. Overall, this is a good paper that points out an aspect of global climate that is typically neglected, but that can have a significant impact. I think the paper deserves to be accepted with minor revisions.

Comments:

1. 1. 53-55 - Note that the main focus of *Abbot and Halevy (2010)* was the effect of aerosol (rather than surface) dust on the snowball climate.
2. Fig. 2 - label the vertical axes in panels (a-d)
3. 1. 85 (also l. 184) - The scaling seems to be with the logarithm of surface erodability, rather than linear in it.
4. 1. 119, figure 2f. - In *Farrell and Abbot (2012)*, we used a simple model to describe a mechanism for a high-dust climate state where atmospheric shortwave absorption by large amounts of dust leads to an inversion and a great reduction in precipitation and therefore an increase in dust emission. Given your model output and the huge dust loadings in Table S1, it seems like you might have found this state in a fancy GCM that includes explicit modeling of the dust cycle. It might be interesting to think more about whether you did in fact find this climate state here, and what the implications of that might be. For example, could the snowball initiation be related to transitioning into or out of this state?
5. 1. 162 - Notice that dust should only make the non-snowball Neoproterozoic climate colder. It should make the snowball Neoproterozoic climate warmer.
6. figure S1 - The caption is messed up.

References

- Abbot, D. S., and I. Halevy (2010), Dust Aerosol Important for Snowball Earth Deglaciation, *Journal of Climate*, 23(15), 4121–4132, DOI: 10.1175/2010JCLI3378.1.
- Farrell, B., and D. Abbot (2012), A mechanism for dust-induced destabilization of glacial climates., *Climate of the Past*, 8(6).

Reviewer #3 (Remarks to the Author):

Comments to the authors and editor: This is an important and innovative paper that should definitely be published. It is the only paper that rigorously addresses Earth's climate for 4 billion years of our planet's history and, therefore, it has very important implications to planetary evolution, evolution of life, evolution of atmospheric systems, and geological feedbacks. I have made numerous small editing changes or recommendations throughout the paper (i.e., "wordsmithing"), which can be accepted or not, as per the wishes of the authors and editor. I have only two substantive issues to mention, one being the citation or acknowledgment of our work on planar sedimentary bedforms in Mesoproterozoic sedimentary rocks of Montana (sorry if this is self-serving – but our work supports the main thesis of this paper; I cited this work in a comment), and the other issue being the possible role of bacterial mats on land surfaces during the Proterozoic (similar to "cryptobiotic" or soil-crust mats of cyanobacteria that cover modern-day arid, desert surfaces in many regions of the world). The Proterozoic was the zenith of cyanobacteria (stromatolites) in the oceans, and land-based bacterial mats were undoubtedly present as well – even in cooler areas (we have these in Montana!). These could have had an important role in actually reducing atmospheric dust to some degree, and should at least be mentioned or considered for "future research." I think the authors should address this issue to avoid having their conclusions dismissed. Overall, an excellent paper!

Response to Reviewers' Comments

Our responses to the reviewers' comments are shown in red below.

Reviewer #1

Overview: This paper points out the fact that many previous paleoclimate modeling efforts have neglected the impact of aerosol dust. More dust aerosol generally leads to cooling, although there are some compensating warming effects of dust described in the paper. The paper describes GCM calculations that show that the cooling due to dust could be as much as 10 K in the global mean. The effect of dust should be greatest before land plants evolved, which makes this work most relevant to the Precambrian. Overall, this is a good paper that points out an aspect of global climate that is typically neglected, but that can have a significant impact. I think the paper deserves to be accepted with minor revisions.

Thanks for the encouragement and the comments below.

Comments:

1. 1. 53-55 - Note that the main focus of *Abbot and Halevy* (2010) was the effect of aerosol (rather than surface) dust on the snowball climate.

Thanks for pointing this out, we overlooked this difference. The text here is now modified to “In both of these studies, however, dust had a warming effect either by lowering the surface albedo by polluting ice¹ or lowering the planetary albedo by floating in the air above glaciated surfaces²”

2. Fig. 2 - label the vertical axes in panels (a-d)

They are labeled now.

3. 1. 85 (also 1. 184) - The scaling seems to be with the logarithm of surface erodability, rather than linear in it.

The scaling here is linear because both the independent variable (k) and dependent variables (e.g. dust emission) described here and displayed in Fig. 2a change logarithmically. It is indeed easy to forget that k is always increased by a factor of 2 instead of by a fixed amount, so a short note is now added in the caption of Fig. 2 to remind the readers about this.

4. 1. 119, figure 2f. - In *Farrell and Abbot* (2012), we used a simple model to describe a mechanism for a high-dust climate state where atmospheric shortwave absorption by large amounts of dust leads to an inversion and a great reduction in precipitation and therefore an increase in dust emission. Given your model output and the huge dust loadings in Table S1, it seems like you might have found this state in a fancy GCM that includes explicit modeling of the dust cycle. It might be interesting to think more about whether you did in fact find this climate state here, and what the implications of that might be. For example, could the snowball initiation be related to transitioning into or out of this state?

Yes, an inversion of global mean atmospheric temperature is found when k is very large, i.e. 0.3; for all other smaller k values, inversion occurs in regions where the dust loading is heavy. However, whether the large decrease in precipitation as shown in Fig. S5 is mainly due to the inversion or the decrease of surface temperature is not so clear for the simulations herein. The precipitation rate decreases by $3.2\%/^{\circ}\text{C}$ when k is increased from 0 to 0.0375, but decreases by $7.5\%/^{\circ}\text{C}$ when k is increased further from 0.0375 to 0.075, and decreases by $11.0\%/^{\circ}\text{C}$ when k is increased again from 0.15 to 0.3. Such decrease is much greater than the Clausius-Clapeyron slope and is probably indicative of the suppression of convective precipitation by the temperature inversion induced by dust, but more detailed analysis in the future is warranted because large change of sea ice area (which could be also reducing precipitation) accompanies the change of global mean surface temperature.

The change in vertical temperature profile could have led to the decrease of the fraction of wet deposition from 32.5% to 30.0% and increase of the fraction of fine dust particles (0-2.5 μm) in the atmospheric dust loading from 45.8% to 56.5%, when k is increased from 0.0375 to 0.3. This change is consistent with but smaller than what was envisaged in *Farrell and Abbot* (2012), where they have speculated that the loading of fine particles in the upper atmosphere would increase much more rapidly than the surface dust emission.

A few sentences have been added to the manuscript near lines 90-96 and 126-132. One extra column is added to Table S1 to show the fraction of fine particles in the total atmospheric dust loading.

We have almost finished the simulations testing the influence of mineral dust on the snowball Earth initiation. The preliminary results show that consideration of dust does not cause the Earth to enter a snowball state at a higher CO_2 level; the

warming effect of the dust by polluting the surface ice and snow almost fully compensate its cooling effect in a near-snowball state. Due to this, we have deleted the second paragraph in the Discussion section of the original manuscript. The results will be presented in another paper, and we will keep in mind the reviewer's suggestion when doing the analyses there.

5.1. 162 - Notice that dust should only make the non-snowball Neoproterozoic climate colder. It should make the snowball Neoproterozoic climate warmer.

Yes, the description is made more specific now (line 174).

6. figure S1 - The caption is messed up.

The caption is now fixed

References

Abbot, D. S., and I. Halevy (2010), Dust Aerosol Important for Snowball Earth Deglaciation, *Journal of Climate*, 23(15), 4121–4132, DOI: 10.1175/2010JCLI3378.1.

Farrell, B., and D. Abbot (2012), A mechanism for dust-induced destabilization of glacial climates., *Climate of the Past*, 8(6).

Reviewer #2

Comments to the authors and editor: This is an important and innovative paper that should definitely be published. It is the only paper that rigorously addresses Earth's climate for 4 billion years of our planet's history and, therefore, it has very important implications to planetary evolution, evolution of life, evolution of atmospheric systems, and geological feedbacks. I have made numerous small editing changes or recommendations throughout the paper (i.e., "wordsmithing"), which can be accepted or not, as per the wishes of the authors and editor. I have only two substantive issues to mention, one being the citation or acknowledgment of our work on planar sedimentary bedforms in Mesoproterozoic sedimentary rocks of Montana (sorry if this is self-serving – but our work supports the main thesis of this paper; I cited this work in a comment), and the other issue being the possible role of bacterial mats on land surfaces during the Proterozoic (similar to "cryptobiotic" or soil-crust mats of cyanobacteria that cover modern-day arid, desert surfaces in many regions of the world). The Proterozoic was the zenith of cyanobacteria (stromatolites) in the oceans, and land-based bacterial mats were undoubtedly present as well – even in cooler areas (we have these in Montana!). These could have had an important role in actually

reducing atmospheric dust to some degree, and should at least be mentioned or considered for “future research.” I think the authors should address this issue to avoid having their conclusions dismissed. Overall, an excellent paper!

Thank you very much for the support of our work, especially the support with observational evidence, which should certainly be cited in the manuscript (near line 105 in the revised manuscript).

The soil crust generation by cyanobacteria cannot be modeled by the current generation of climate models, and is indeed an uncertainty that we should discuss. Biogenic soil crust develops when the region is not hyperarid and little disturbed (Belnap and Longe, 2001; personal communication with J. Belnap and Y. Ashkenazy). If the soil crust is very well developed, i.e., the crust is thick and organic concentration in the soil is high, it can even prohibit dust emission. The soil crust should be present during the Precambrian where wind is weak and precipitation is moderate or heavy, but its detailed distribution is unknown. Given the fact that strong surface dust emission and deposition always occur in present day and during the Earth’s history (e.g. Brookfield and Ahlbrandt, 1983; the reviewer likely knows these observations much better than us), we think it might not be a key factor in controlling global dust emission, and the uncertainty might have been contained in the range of surface erodibility (k) we have tested. In the future, it will be worthwhile to model the dynamics of soil crust explicitly and its influence on global dust emission and dust loading during the Precambrian.

A paragraph is added in Discussion section basically describing the point above.

It is believed that large cratonic land areas came into being in the Late Archean and Paleoproterozoic with the supercontinents Kenorland (also known as Superia) and Columbia/Nuna, respectively.

We have found two recent studies (Bindeman et al., 2018; Johnson and Wing, 2020), both supporting that the area of continents was likely close to that of present day since 2.5 billion years ago. These are added to the Discussion section of the manuscript.

Why are the orbital parameters for 1990 used? Why this specific year?

By default, the orbital parameters for year 1990 are prescribed in the climate model we used. Since our major focus is to test the influence of land plants on dust emission, the orbital parameters are not changed. We do not believe the orbital parameters will affect the results substantially, but it will be useful to do some sensitivity tests when the computational cost becomes affordable.

References

Belnap J, Lange OL (eds). *Biological Soil Crusts: Structure, Function, and Management*, Ecological Studies, 150, Springer: Berlin, Heidelberg, New York, 2001, 503p

Bindeman IN, Zakharov DO, Palandri J, Greber ND, Dauphas N, Retallack GJ, *et al.* Rapid emergence of subaerial landmasses and onset of a modern hydrologic cycle 2.5 billion years ago. *Nature* 2018, **557**(7706): 545-+.

Brookfield, ME, Ahlbrandt TS (eds). *Eolian Sediments and Processes*, Developments in Sedimentology, 38, Elsevier, Amsterdam, 1983, 660p.

Johnson BW, Wing BA. Limited Archaean continental emergence reflected in an early Archaean O-18-enriched ocean. *Nat Geosci* 2020, **13**(3): 243-+.

REVIEWERS' COMMENTS:

Reviewer #1 (Remarks to the Author):

The authors have addressed my comments and the paper is ready for publication.

Response to Reviewers' Comments

Because there are no further comments or suggestions received from the reviewers, there is correspondingly no response provided.

We have revised the manuscript according to the editorial comments received, and the changes are tracked in the submitted manuscript file.